# Aging Characteristics and Fate Analysis of Liquid Digestate Ammonium Nitrogen Disposal in Farmland Soil

**Zichen Wang** [1,2,3], **Guofeng Sun** [1,3], **Liping Zhang** [1,3], **Wei Zhou** [1,3], **Jing Sheng** [1,3], **Xiaomei Ye** [3,*], **Ademola O. Olaniran** [2,*], **Evariste B. Gueguim Kana** [2] **and Hongbo Shao** [1]

1    Institute of Agricultural Resources and Environment, Jiangsu Academy of Agricultural Sciences, Nanjing 210014, China
2    Discipline of Microbiology, School of Life Sciences, University of KwaZulu-Natal, Pietermaritzburg 4000, South Africa
3    Key Laboratory of Crop and Livestock Integrated Farming, Ministry of Agriculture and Rural Affairs, Nanjing 210014, China
*    Correspondence: yexiaomei610@126.com (X.Y.); olanirana@ukzn.ac.za (A.O.O.)

**Abstract:** Water environment safety is the focus of engineering measures to eliminate liquid digestate in farmland. It is of great significance to study the aging characteristics of soil absorbing and fate of liquid digestate ammonium nitrogen ($NH_4^+$-N) to realize safe and efficient disposal. In this paper, simulation experiments of digesting $NH_4^+$-N (with application of 0, 120, 180, and 300 kg/hm$^2$) by static soil column are carried out to study disposal efficiency, migration and transformation characteristics, and fate proportion of $NH_4^+$-N in saturated water content soil. The result showed that after 3 days of application, the overlying water $NH_4^+$-N concentration decreased by 63.5–80.7%, and the reduction rate of total $NH_4^+$-N was 65.8–82.3%. After 4 days, the $NH_4^+$-N concentration of pore water in the 0–10 cm soil layer reached the peak value. After 7 days, the $NH_4^+$-N concentration adsorbed by the 0–10 cm soil layer reached the peak value. After 15 days, the overlying water $NH_4^+$-N concentration decreased by 97.0–98.7%, the reduction rate was 97.9–99.2%, and the proportion of $NH_4^+$-N absorbed in the 0–10 cm soil layer accounted for 63.5–76.3%. The disposal is mainly based on soil sorption and pore water migration. A duration of 0–3 days is the rapid disposal period, and 15 days is the completion period of safe digestion.

**Keywords:** waste biomass utilization; liquid digestate; ammonium nitrogen; sorption; migration; transformation

## 1. Introduction

To reduce the environmental pollution of the livestock and poultry breeding industry [1], in recent years, the Chinese government has continued to strengthen the construction of biogas projects in livestock and poultry farms, and has continuously promoted the transformation and upgrading of rural biogas projects in combination with the green development of agriculture and the action of replacing chemical fertilizers with organic fertilizers [2–4], using the anaerobic fermentation process to dispose of aquaculture excrement, in order to realize the harmless treatment and resource reuse of aquaculture manure [5,6]. By the end of the year 2020, 128,976 small and medium-sized biogas projects and 10,122 large-scale biogas projects have been built nationwide [7]. Liquid digestate is a by-product of biogas engineering, accounting for more than 90% of the total fermentation residue [8]. According to estimates, China produces 1.12 billion tons of liquid digestate annually [7]. Due to the large amount of liquid digestate produced, high storage and transportation costs, difficult treatment to meet standards, and low commercialization value, there are serious secondary pollution environmental risks [9]. The treatment and utilization of liquid digestate have become the focus and difficulty of domestic and foreign research [10–15]. The use of farmland soil, crops, and microorganisms living in the soil to absorb liquid



digestate is a widely recognized green treatment method [16,17], but the amount of farmland consumption cannot exceed the limit of land carrying capacity [18]; otherwise, it will cause serious pollution to the surrounding soil and water bodies [19,20]. High ammonia nitrogen ($NH_4^+$-N) concentration in liquid digestate components is the primary risk factor for environmental pollution [16,21,22]. Therefore, it is of great significance to study the aging characteristics of farmland soil to absorb liquid digestate and the fate of $NH_4^+$-N to realize safe and efficient disposal of liquid digestate in farmland.

According to the soil nitrogen transport theory [23,24], after the liquid digestate is applied to the farmland, $NH_4^+$-N in the unsaturated water content soil completes vertical and horizontal transport with water in convection mode, while the saturated water content soil completes migration from the high-concentration area to the low-concentration area by diffusion infiltration. During the migration process, $NH_4^+$-N will be rapidly adsorbed and gradually nitrified into nitrate nitrogen ($NO_3^-$-N) [25]. When local surface runoff is generated, $NH_4^+$-N and $NO_3^-$-N not adsorbed by soil particles will be lost and leached with water at the same time, thus polluting the surrounding water sources [26–28]. Liquid digestate contains relatively more available nitrogen. Therefore, it has been proposed that liquid digestate application will lead to more nitrogen leaching loss than manure application. However, after a two-and-a-half-year corn field experiment, there was no significant difference in nitrogen leaching amount between digestate application and manure and chemical fertilizer application [29]. When liquid digestate is applied by spraying and deepening in the slack season in autumn, there is no risk of $NH_4^+$-N and $NO_3^-$-N leaching. However, when liquid digestate is applied by injection, there is still a potential risk of $NH_4^+$-N leaching even when the nitrogen dosage of liquid digestate is 90 kg/(hm²·d) [30]. The study on disposal of liquid digestate in paddy fields shows that the concentration of $NH_4^+$-N in field surface water decreases rapidly 1–4 days after application [16,31]. After 8 days of application, the $NH_4^+$-N concentration in field surface water can basically reach the level of the blank control field. The concentration of $NH_4^+$-N in groundwater is always lower than that of chemical fertilizer treatment, and does not increase with the increase in liquid digestate application amount [32]; meanwhile, the content of $NO_3^-$-N in field surface water and groundwater will not increase significantly [33]. The increase in ammonia volatilization is considered to be the main negative impact of liquid digestate application on the farmland environment [31,34,35]. After liquid digestate application, the ammonia volatilization is higher than that of the total chemical fertilizer treatment. With the increase in liquid digestate dosage, the ammonia volatilization is increased, and the soil wetting or flooding conditions can reduce the ammonia volatilization [36–38].

The above studies monitored and qualitatively analyzed the changes of nitrogen concentration in farmland water after liquid digestate was applied, but there was a lack of quantitative research on the reduction process of liquid digestate $NH_4^+$-N. It is a new way to treat liquid digestate by using farmland with saturated water content for disposal, which is different from the fertilizer utilization of liquid digestate. When taking measures to absorb liquid digestate in farmland with saturated water content, farmers are more concerned about the main destination of $NH_4^+$-N after liquid digestate is applied in farmland and the time required for the discharge of field water quality up to standard. In this paper, the simulation experiment of indoor static soil column is used to study the time-effect of absorbing $NH_4^+$-N from liquid digestate in saturated water content soil, analyze the migration and transformation characteristics and fate ratio of $NH_4^+$-N from liquid digestate, and discuss the time required for the discharge of field surface water quality to meet the standard, so as to provide theoretical basis and technical guidance for the efficient elimination of liquid digestate in farmland under the safety of water environment.

## 2. Materials and Methods

### 2.1. Materials

The tested liquid digestate was taken from Jiangsu Yangyu Ecological Agriculture Co., Ltd. (Taizhou, China), which produces around 120,000 commercial pigs annually, and

was recognized by the Ministry of Agriculture and Rural Affairs as a "pig standardization demonstration farm", a provincial key leading enterprise of agricultural industrialization in Jiangsu Province, and a comprehensive demonstration base of ecological circular agriculture of Jiangsu Academy of Agricultural Sciences (Nanjing, China). The liquid digestate was generated from liquid manure and sewage through primary anaerobic fermentation in biogas engineering with a continuous stirred-tank reactor (CSTR) and a secondary anaerobic fermentation using a covered lagoon storage process (Figure 1). Once taken back to the laboratory, the liquid digestate was stored in sealed plastic barrels and was mixed well. The solid and insoluble matter were filtered out of digestate through a 0.25-mm mesh screen and were not used for the experiment. The average properties of liquid digestate measured before this test are: pH value $8.05 \pm 0.06$, total nitrogen (TN) $461.63 \pm 5.39$ mg/L, $NH_4^+$-N $409.12 \pm 6.75$ mg/L, $NO_3^-$-N $31.56 \pm 0.08$ mg/L, total phosphorus (TP) $17.72 \pm 0.14$ mg/L, total potassium (TK) $279 \pm 2.74$ mg/L, chemical oxygen demand (COD) $470.11 \pm 7.85$ mg/L, electrical conductivity (EC) $3.81 \pm 0.01$ mS/cm, and total solid (TS) $1.63 \pm 0.01$ g/L.

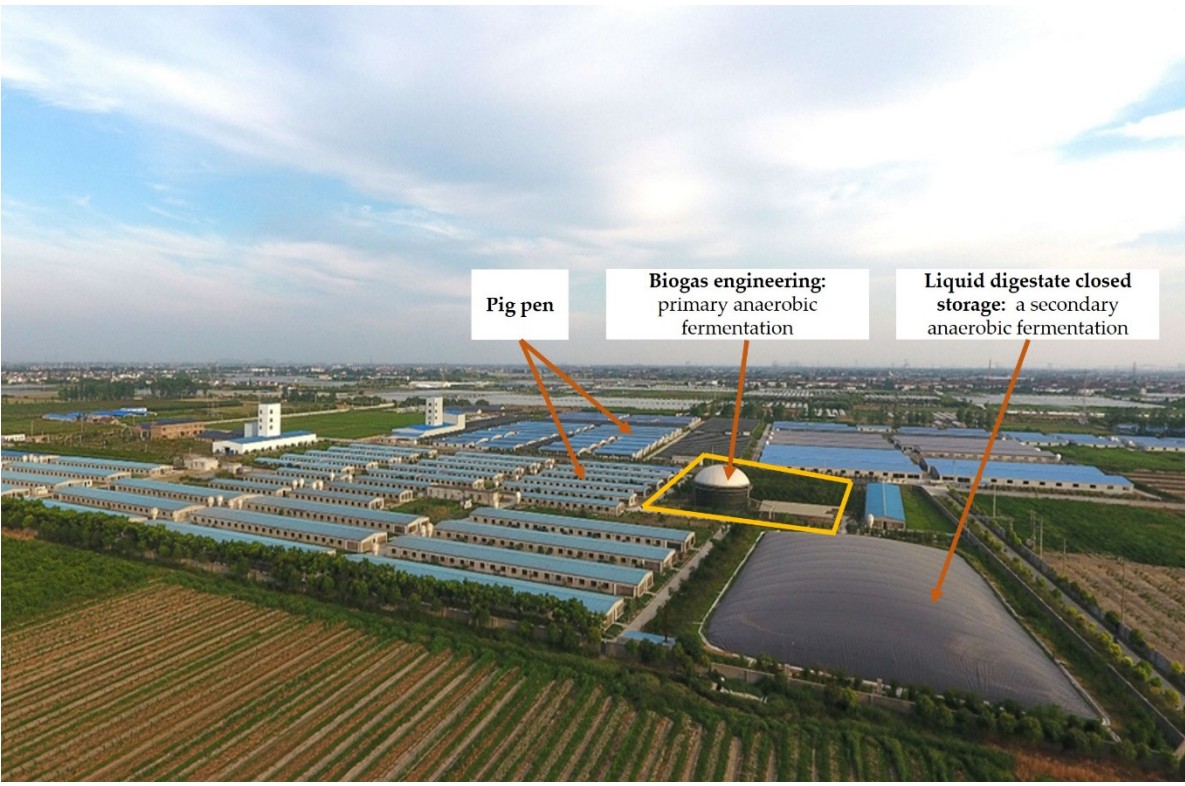

**Figure 1.** Biogas engineering and liquid digestate storage facilities in pig farms.

The experimental soil was collected from the Xinbei District of Changzhou City in the Yangtze River Basin of China. It was 0–20 cm topsoil of permanent basic farmland, and its texture was silty loam. The soil was dried naturally, while stones, plant roots, and other sundries found in the soil were removed. The soil was then crushed with a round wooden stick, sieved through a 2 mm aperture mesh screen, and finally fully mixed into a clean plastic storage box for future use. The basic physical and chemical properties of the soil are: soil organic matter (SOM) $29.09 \pm 0.39$ g/kg, pH value $6.45 \pm 0.04$, TN $1.16 \pm 0.17$ g/kg, $NH_4^+$-N $8.93 \pm 0.57$ mg/kg, $NO_3^-$-N $56.97 \pm 0.43$ mg/kg, TP $0.57 \pm 0.02$ g/kg, available phosphorus (AP) $13.10 \pm 1.47$ mg/kg, available potassium (AK) $122.91 \pm 13.21$ mg/kg, cation exchange capacity (CEC) $16.27 \pm 0.49$ cmol/kg, and EC $492.67 \pm 19.14$ μS/cm. The soil particle group is composed of 30.5% particles with a particle size of 2–0.05 mm, 52.9% particles with a particle size of 0.05–0.002 mm, and 16.5% particles with a particle size less than 0.002 mm.

### 2.2. Static Soil Column Fabrication

An indoor static soil column was used to simulate the experiment (Figure 2). The manufacturing method for the soil column is as follows: firstly, a flat-bottom glass tube with an inner diameter of 6.0 cm and a height of 30.0 cm is customized, and the cross-sectional surface area of the test tube is 28.26 cm$^2$. Use a 1% electronic balance to accurately weigh 600 g of the prepared soil into a flat-bottomed tube, shake the tube to make the soil solid (the soil depth is about 20 cm), and then add 369.9 mL of deionized water (the data is the sum of the saturated water content and pore water content of the soil used in the actual test) so that the water content of the soil column reaches the maximum saturated state, and stand for use after standing overnight.

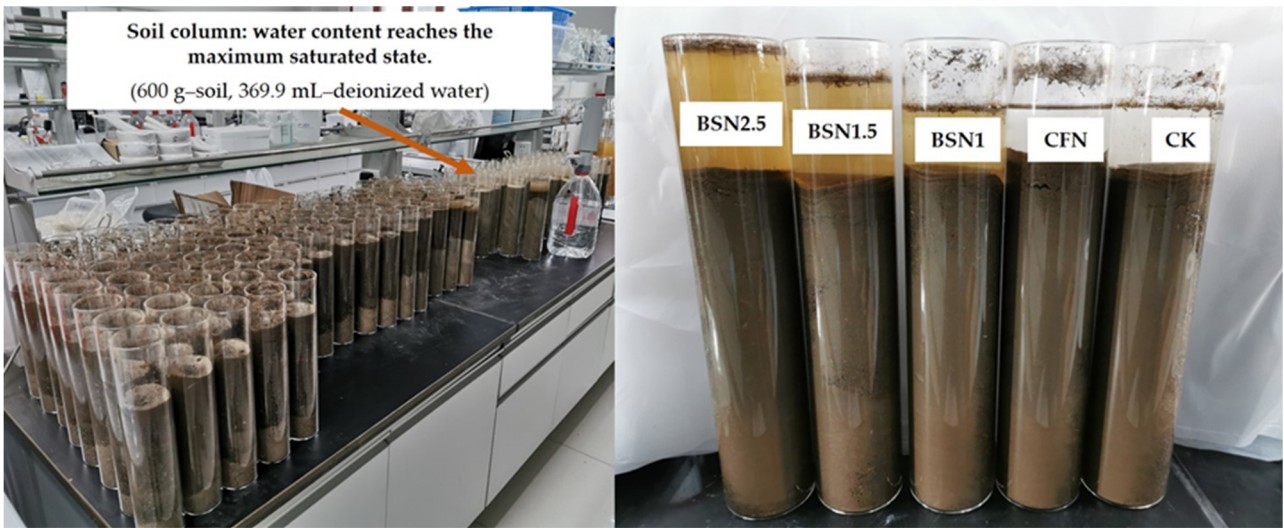

**Figure 2.** Physical photos of static soil column.

### 2.3. Design and Setting

There are 5 treatments in the experiment as follows:

Treatment ①: Apply chemical fertilizer $NH_4^+$-N 120 kg/hm$^2$. The amount refers to the customary nitrogen fertilizer amount of farmers in the rice panicle fertilizer stage of saturated water content paddy fields, which is recorded as: CFN1. Weigh the analytical pure reagent $NH_4Cl$ and add it into deionized water, then prepare 409 mg/L $NH_4^+$-N solution with the same concentration as the liquid digestate. Measure 82.9 mL of solution, and add it to the soil column surface.

Treatment ②: Apply liquid digestate $NH_4^+$-N 120 kg/hm$^2$, which is 1 times the amount of chemical fertilizer $NH_4^+$-N in Treatment ①, denoted as: BSN1. Measure 82.9 mL of liquid digestate, and add it to the surface of the soil column.

Treatment ③: Apply liquid digestate $NH_4^+$-N 180 kg/hm$^2$, which is 1.5 times the amount of chemical fertilizer $NH_4^+$-N in Treatment ①, referring to the accustomed nitrogen fertilizer dosage of farmers in the rice base-tiller fertilizer period of saturated water content paddy fields, denoted as: BSN1.5. Measure 124.4 mL of liquid digestate, and add it to the surface of the soil column.

Treatment ④: Apply liquid digestate $NH_4^+$-N 300 kg/hm$^2$, which is 2.5 times the amount of chemical fertilizer $NH_4^+$-N in Treatment ①, with reference to the total nitrogen fertilizer dosage used by farmers in the rice season in paddy fields with saturated water content, recorded as: BSN2.5. Measure 207.3 mL of liquid digestate, and add it to the surface of the soil column.

Treatment ⑤: No fertilization control, keep the same amount of water as Treatment ①, denoted as: CK. Measure 82.9 mL of deionized water, and add it to the soil column surface.

Thirty replicates were set up for each treatment, for a total of 150 soil pillars. Take destructive sampling, take 3 repeated soil columns each time, and discard the soil columns after the measurement.

*2.4. Sampling and Analysis*

At 0, 1, 2, 3, 4, 5, 7, 9, 12, and 15 days after the application of liquid digestate, the overlying water was taken from the soil column, and the concentrations of $NH_4^+$-N and $NO_3^-$-N in the overlying water were measured. After removing the overlying water, excavate 0–10 cm topsoil in the soil column, centrifuge at 4000 r/min for 5 min, and take the supernatant (soil pore water) to measure $NH_4^+$-N and $NO_3^-$-N concentrations. The soil after centrifugation (Soil Sample 1) was retained, and the soil water content, soil water-soluble $NH_4^+$-N content [39], and soil ion-exchanged $NH_4^+$-N content [40] were determined.

Determination method of soil water-soluble $NH_4^+$-N content [41]: Take 8.00 g of the centrifuged soil (Soil Sample 1) sample, put it into a centrifuge tube, add 40 mL of deionized water according to the solid–liquid ratio of 1:5, and tighten the sealing cap of the centrifuge tube. Mix thoroughly, shake at 160 r/min for 30 min at 25 °C with a thermostatic oscillator, then centrifuge at 4000 r/min for 20 min, collect the supernatant, and repeat the above operation twice for the soil samples in the centrifuge tube. The supernatants collected three times were mixed for the determination of soil water-soluble $NH_4^+$-N content. Retain the centrifuge tube and the soil in the tube (Soil Sample 2).

Determination method of ion-exchanged $NH_4^+$-N content [41]: Add 40 mL of KCl solution with a concentration of 0.5 mol/L to the centrifuge tube where the soil (Soil Sample 2) after extraction of water-soluble $NH_4^+$-N is located, tighten the sealing cap of the centrifuge tube, and mix well. At 25 °C, shake at 160 r/min for 60 min with a thermostatic oscillator, then centrifuge at 4000 r/min for 10 min, collect the supernatant, and repeat the above operation twice for the soil samples in the centrifuge tube. The supernatants collected three times were mixed and used to determine the ion-exchanged $NH_4^+$-N content of the soil.

The $NH_4^+$-N and $NO_3^-$-N contents of all water quality in this experiment were determined by a SKALAR SAN++ full-automatic flow analyzer (Skalar Analytical B.V. Products, Breda, The Netherlands). Daily water evaporation loss of the soil column was measured by using a 1% electronic balance to weigh and calculate the difference with the subtraction method.

*2.5. Calculation Formula*

Water evaporation loss of soil column:

$$V_t = \frac{(m_0 - m_t)}{\rho} \tag{1}$$

In the Formula (1): $V_t$ is liquid digestate evaporation loss of overburden water in t day (mL); $m_0$ is overall mass of soil column on Day 0 (within 8 h) after liquid digestate is applied (g); $m_t$ is the overall mass of soil column on $t$-day (g); $\rho$ is density of water (g/mL).

Reduction rate of liquid digestate $NH_4^+$-N in overlying water:

$$R(\%) = \frac{M - C_t \cdot (V_0 - V_t)}{M} \times 100 \tag{2}$$

In Formula (2): $M$ is total application amount of $NH_4^+$-N (mg); $C_t$ is $NH_4^+$-N concentration in overlying water on $t$-day (mg/L); $V_0$ is initial application volume of liquid digestate (L); $V_t$ is liquid digestate evaporation loss on t-day (L).

Fate of liquid digestate $NH_4^+$-N:

$$F(\%) = \frac{C_t \cdot V}{M} \times 100 = \frac{\omega_t \cdot m}{M} \times 100 \tag{3}$$

In Formula (3): $M$ is total application amount of $NH_4^+$-N (mg); $C_t$ is $NH_4^+$-N, $NO_3^-$-N concentration in overlying water or in soil pore water on $t$-day (mg/L); $V$ is residual volume of overlying water or pore water volume (L); $\omega_t$ is the concentration of $NH_4^+$-N and $NO_3^-$-N adsorbed by the soil on t-day (mg/kg); $m$ is soil mass (kg).

### 2.6. Data Analysis

Microsoft® Excel® 2016 MSO (16.0.4549.1001) 64-bit was used for the summary, analysis, and graphing of experimental data, and IBM SPSS Statistics (22) software was used for one-way ANOVA and Duncan's method for analysis of variance and multiple comparisons ($\alpha = 0.05$). Data in the graph are mean ± standard deviation.

## 3. Results

### 3.1. Variation Characteristics of $NH_4^+$-N Concentration and $NH_4^+$-N Reduction Rate in Overlying Water

The dynamic change process of liquid digestate $NH_4^+$-N consumption and the reduction rate in overlying water with saturated water content farmland soil are shown in Figure 3. After liquid digestate was applied to the soil surface with saturated water content, the concentration of $NH_4^+$-N in the overlying water decreased gradually with the extension of time, and the reduction rate of the total amount of $NH_4^+$-N gradually increased. Among them, 0–3 days is the rapid digestion period. During this period, the concentration of $NH_4^+$-N in the overlying water drops rapidly, the decline range is 63.5–80.7% (Figure 3a), and the total reduction rate of $NH_4^+$-N reaches 65.8–82.3% (Figure 3b).

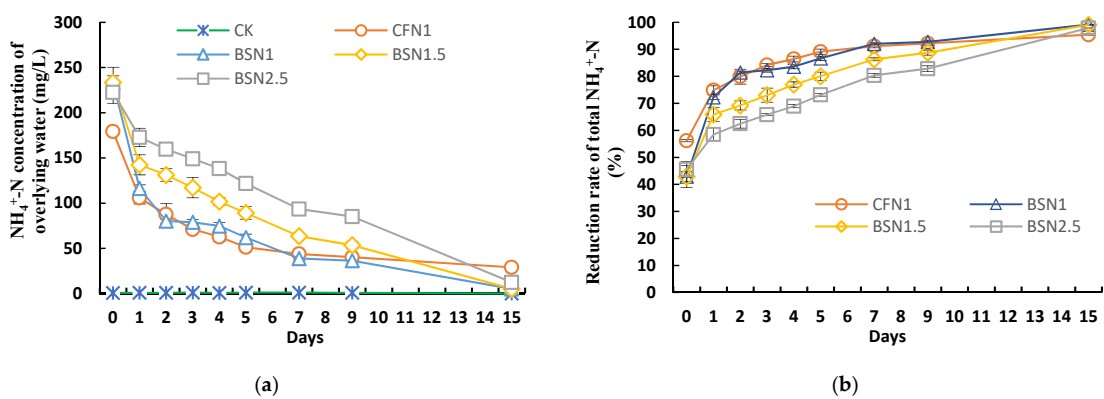

**Figure 3.** Digestion characteristics (**a**) and reduction rate (**b**) of liquid digestate $NH_4^+$-N in overlying water.

Compared with CFN1 treatment, under the condition of equal $NH_4^+$-N, the $NH_4^+$-N concentration in overlying water of BSN1 treatment increased significantly ($p < 0.05$) and then decreased rapidly after 0 days of liquid digestate application (the sampling time was within 8 h after liquid digestate application). On the third day after application, the $NH_4^+$-N concentration of the overlying water decreased to 78.88 mg/L, which was lower than the discharge concentration of 80 mg/L, specified in the emission standard of pollutants for the livestock and poultry breeding industry (GB18596-2001) [42], and lower than that of the CFN1 treatment, but the difference was not significant. The $NH_4^+$-N concentration rebounded slightly after application for 3–7 days and remained lower than that of CFN1 treatment after application for 7 days. After 15 days, the $NH_4^+$-N concentration in the overlying water of BSN1 treatment decreased to 5.16 mg/L, which was significantly lower than that of CFN1 treatment ($p < 0.05$). The $NH_4^+$-N concentration decreased by 98.7%, and the total reduction rate of $NH_4^+$-N reached 99.2%. However, with high ammonium nitrogen treatment of BSN1.5 and BSN2.5, the $NH_4^+$-N concentration in the overlying water was significantly higher than that of CFN1 treatment from 0 to 9 days after application, but after 15 days, the $NH_4^+$-N concentration decreased to 5.18 mg/L and 12.32 mg/L,

respectively, which were significantly lower than that of CFN1 treatment ($p < 0.05$). The $NH_4^+$-N concentration of the two treatments decreased by 98.7% and 97.0%, and the total reduction rate of $NH_4^+$-N was 99.0% and 97.9%.

### 3.2. Migration and Soil Sorption Characteristics of Liquid Digestate $NH_4^+$-N

Figure 4a shows the change of $NH_4^+$-N concentration in soil pore water in the 0–10 cm soil layer. After the application of liquid digestate, the $NH_4^+$-N in the overlying water diffused and migrated downward, and the $NH_4^+$-N concentration in soil pore water increased rapidly. The larger the amount of liquid digestate applied, the higher the $NH_4^+$-N concentration in soil pore water in the 0–10 cm soil layer. On the fourth day after application, the $NH_4^+$-N content in the pore water reached a peak value and then decreased slowly. The $NH_4^+$-N concentration of BSN1, BSN1.5, and BSN2.5 treatments were 24.41 mg/L, 27.40 mg/L, and 28.91 mg/L, which were significantly higher than those of CFN1 treatment by 16.4%, 30.7%, and 37.9%, respectively ($p < 0.05$).

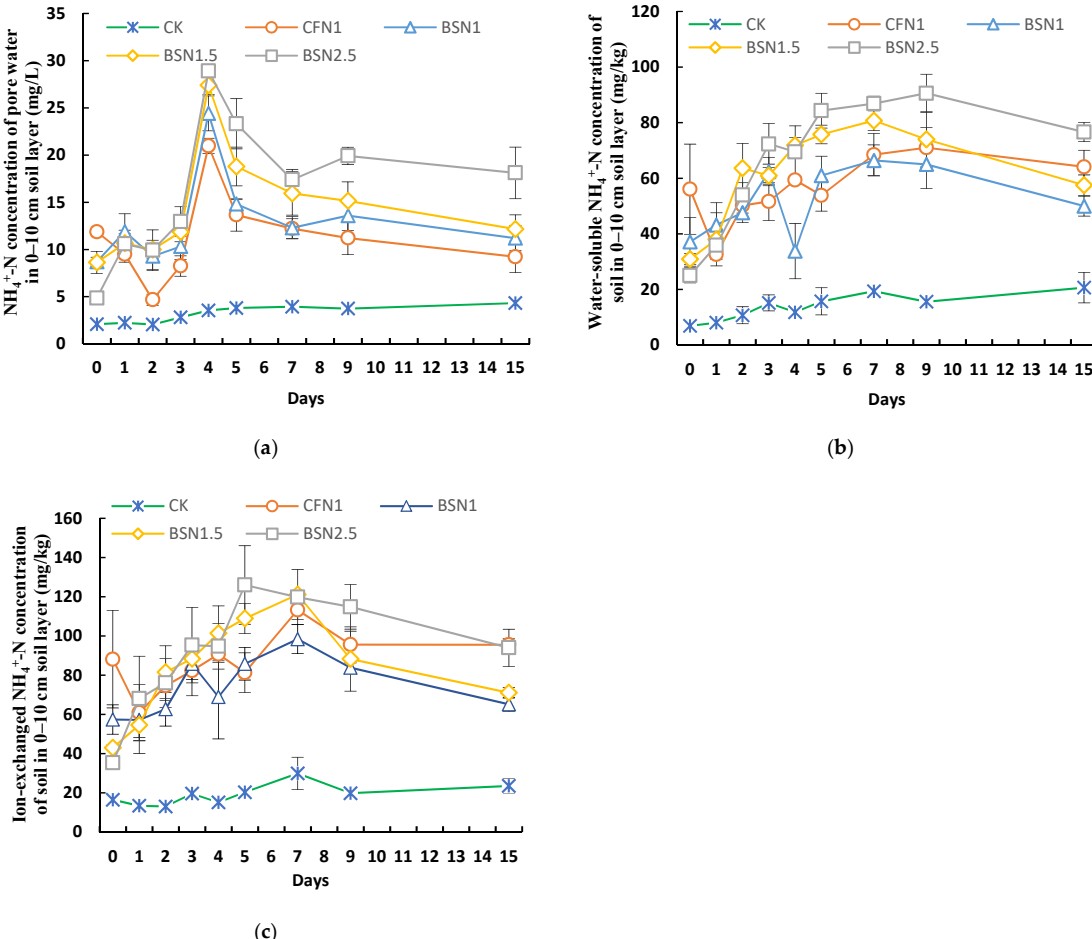

**Figure 4.** Migration of $NH_4^+$-N in soil pore water (**a**) and changes of $NH_4^+$-N adsorbed by soil (**b**,**c**).

Figure 4b,c shows the changes of soil water-soluble $NH_4^+$-N and ion-exchanged $NH_4^+$-N in the 0–10 cm soil layer. After the liquid digestate was applied, the increase in $NH_4^+$-N concentration in soil pore water created a good environment for soil particles to adsorb $NH_4^+$-N. The soil water-soluble $NH_4^+$-N concentration and ion-exchange $NH_4^+$-N concentration increased first and then decreased, and the more liquid digestate applied, the higher the soil water-soluble $NH_4^+$-N concentration and the ion-exchanged $NH_4^+$-N concentration. Under the condition of equal $NH_4^+$-N content, the soil water-soluble $NH_4^+$-N concentration and ion-exchange $NH_4^+$-N concentration of BSN1 treatment reached the peak value on

the seventh day after liquid digestate application, but they were 3.0% and 13.1% lower than those of CFN1 treatment respectively, and the difference between treatments was not significant. With the high ammonium nitrogen content BSN1.5 treatment, on the seventh day, the soil water-soluble $NH_4^+$-N concentration and ion-exchange $NH_4^+$-N concentration reached the peak, which were significantly higher than that of CFN1 treatment by 17.9% and 7.0%, respectively ($p < 0.05$). The concentration of ion-exchange $NH_4^+$-N in BSN2.5 treatment reached the peak on the fifth day, but the concentration of water-soluble $NH_4^+$-N reached the peak on the ninth day, which was significantly higher than that in BSN1 treatment. After the $NH_4^+$-N sorption [43] reached the peak, desorption and transformation gradually appeared. On the 15th day, the concentration of water-soluble $NH_4^+$-N and the concentration of ion-exchange $NH_4^+$-N in BSN1 and BSN1.5 treatments were lower than that in CFN1 treatment, and the difference of ion-exchange $NH_4^+$-N concentration reached a significant level ($p < 0.05$).

### 3.3. Characteristics of Liquid Digestate $NH_4^+$-N Converted to $NO_3^-$-N

The concentration change of liquid digestate $NH_4^+$-N converted to $NO_3^-$-N was shown in Figure 5. The change trends of $NO_3^-$-N in overlying water (Figure 5a), pore water (Figure 5b), and soil water-soluble (Figure 5c) are basically the same. $NO_3^-$-N concentrations were consistently low and there were no significant differences between treatments. Since the seventh day, the $NO_3^-$-N concentrations in overlying water, pore water, and soil water-soluble $NO_3^-$-N of treatments BSN1, BSN1.5, and BSN2.5 all increased significantly compared with those of CFN1 and CK treatments ($p < 0.05$). The greater the amount of liquid digestate application, the greater the increase in $NO_3^-$-N.

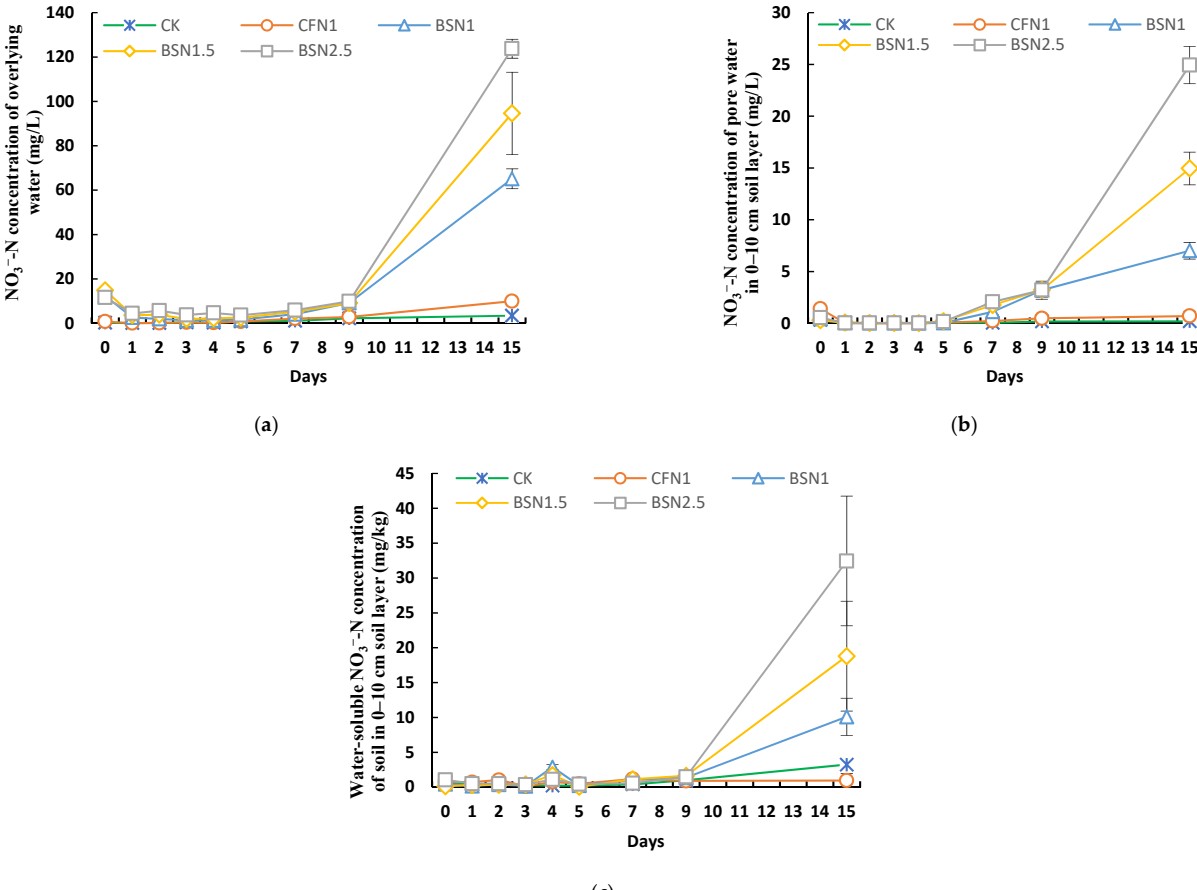

**Figure 5.** Concentration changes of $NO_3^-$-N in overlying water (**a**), pore water (**b**), and soil water-soluble (**c**).

*3.4. Fate Analysis of Farmland Absorbing Liquid Digestate $NH_4^+$-N*

Figure 6 is the analysis of the fate of $NH_4^+$-N in BSN1, BSN1.5, and BSN2.5 treatments on the 15th day of liquid digestate application. The disposal of liquid digestate $NH_4^+$-N is mainly based on soil particle sorption and conversion, but with the increase in liquid digestate application, the proportion of $NH_4^+$-N absorbed by soil decreases. The proportions of nitrogen (sum of $NH_4^+$-N and $NO_3^-$-N) in soil sorption and pore water storage in the 0–10 cm soil layer of treatments BSN1, BSN1.5, and BSN2.5 accounted for 76.3%, 67.1%, and 63.5% of the total applied $NH_4^+$-N, respectively. The residual proportions of overlying water were 0.9%, 1.1%, and 2.4%, respectively, and the proportions of other destinations (including migration to deeper soil layers, transformation, and volatilization loss of overlying water, etc.) were 22.8%, 31.8%, and 34.1%, respectively. Among them, BSN1, BSN1.5, and BSN2.5 treatments accounted for 62.8%, 49.7%, and 44.7% of the total applied $NH_4^+$-N by soil sorption in the 0–10 cm soil layer, respectively, and the $NH_4^+$-N adsorbed by soil ion-exchange state was greater than that adsorbed by water-soluble state. The proportion of nitrogen contained in pore water is 7.5%, 8.3%, and 8.4%, which are higher than the corresponding residual amount of overlying water, indicating that the diffusion and migration of liquid digestate $NH_4^+$-N from overlying water to interstitial water in the 0–10 cm soil layer has been completed after 15 days of liquid digestate application, and the more application, the more migration. The proportion of water-soluble $NO_3^-$-N in soil accounts for 6.1%, 9.2%, and 10.3%, indicating that the liquid digestate $NH_4^+$-N has been transformed into $NO_3^-$-N in the 0–10 cm soil layer after 15 days of application, and the more amount of liquid digestate applied, the greater the quantity of the transformation.

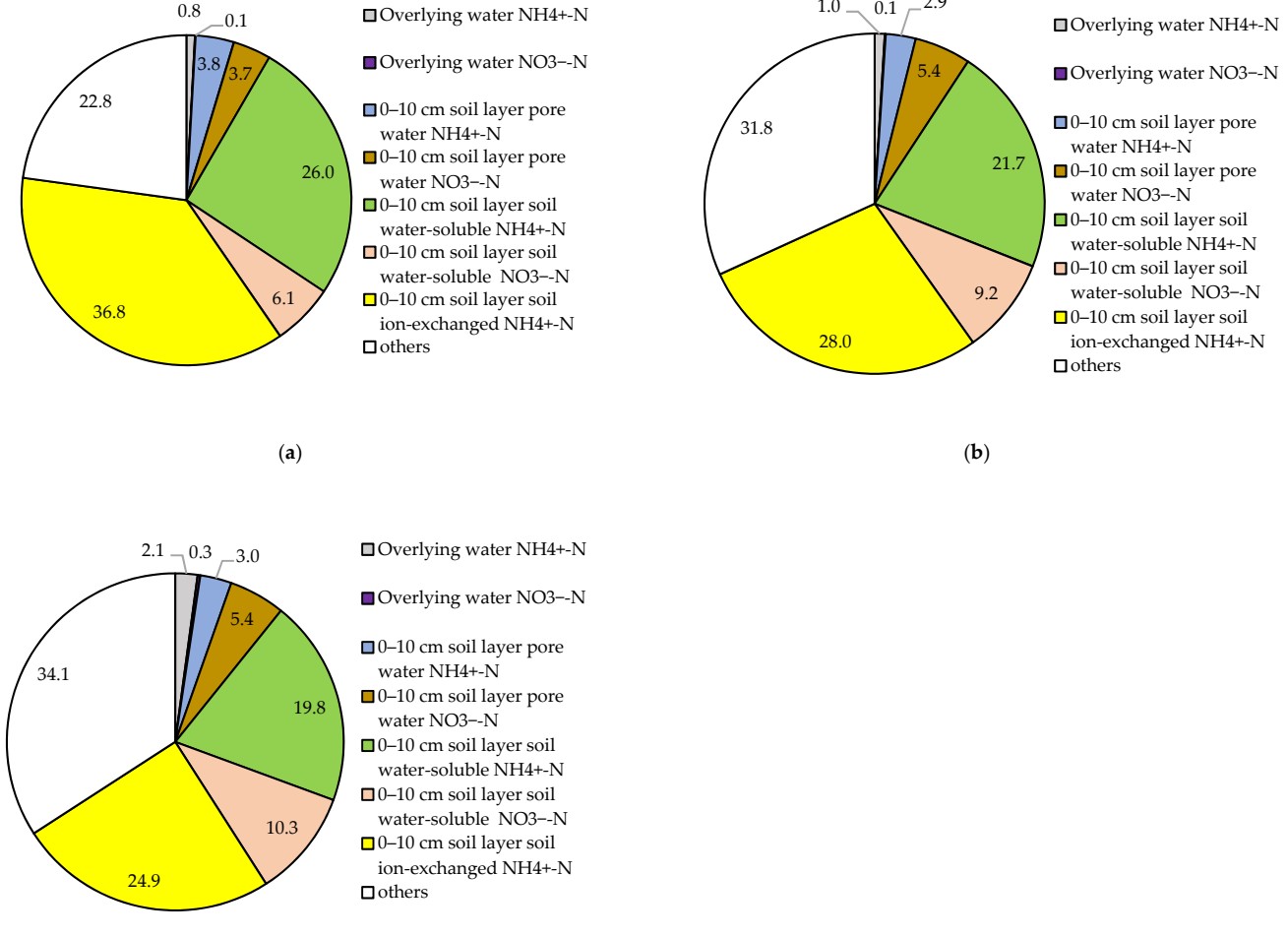

**Figure 6.** Fate analysis of liquid digestate $NH_4^+$-N in treatments of BSN1 (**a**), BSN1.5 (**b**), and BSN2.5 (**c**).

## 4. Discussions

### 4.1. Time Node of Liquid Digestate $NH_4^+$-N Removal in Farmland

Saturated water content farmland has the self-regulation function of the soil-microbial complex system and the comprehensive purification ability of pollutants, and has great liquid digestate disposal potential. The safe bearing capacity of farmland soil and the frequency of digestion [32,44] are important parameters for determining the area configuration of liquid digestate in land-consumption farms in the combined planting and breeding system. Research on the engineering measures for disposal of liquid digestate in paddy fields shows that the $NH_4^+$-N concentration in field water is affected by the digested amount of liquid digestate. At the beginning of application, it increased with the increase in liquid digestate and decreased significantly with time. The $NH_4^+$-N concentration decreased by 47.52–85.27% after 3 days of liquid digestate application [16], and the concentration of $NH_4^+$-N in field water is stably lower than the emission concentration of 80 mg/L specified in the emission standard of pollutants for the livestock and poultry breeding industry (GB18596-2001). It can be stably lower than 40 mg/L after 5 days of application [31]. The results of this study are slightly different from the above reports in the digestion time of $NH_4^+$-N in liquid digestate. After 3 days of application, only the $NH_4^+$-N concentration in the overlying water of BSN1 treatment decreased to less than 80 mg/L. After 7 days

of application, BSN1 and BSN1.5 treatments can stabilize below 80 mg/L, of which BSN1 treatment is lower than 40 mg/L. The reason might be that the static soil column used in this study is a soil-microbial composite system, which lacks the participation of farmland plants, so the digestion speed and aging are slightly delayed.

In addition, by monitoring the change of $NH_4^+$-N concentration in field water, we can predict the water environment pollution risk and water quality standard discharge time node of farmland disposal liquid digestate engineering measures [16,31,45,46]. However, it is impossible to distinguish whether the main reason for the decrease in $NH_4^+$-N concentration is farmland digestion or farmland irrigation water dilution, which has been questioned in production practice. This study further quantifies the reduction rate of $NH_4^+$-N in the liquid digestate. From the perspective of reduction of $NH_4^+$-N input, it is verified again that the first three days after liquid digestate application is a rapid reduction period, during which the prohibition of runoff plays an important role in preventing environmental pollution of surrounding water bodies. After 15 days of application, under the condition of an equal amount of $NH_4^+$-N, the total amount of $NH_4^+$-N in the overlying water of liquid digestate treatment decreased by 99.2%, significantly lower than that of fertilizer $NH_4^+$-N treatment, which proved that the purpose of disposal liquid digestate $NH_4^+$-N had been achieved, and 15 days could be used as the time node for the end of the first digestion cycle.

### 4.2. Migration and Transformation Characteristics of $NH_4^+$-N in Farmland Consuming Liquid Digestate

In this study, under the condition of applying the same amount of $NH_4^+$-N, the $NH_4^+$-N concentration in the overlying water of the liquid digestate treatment was lower than that of the chemical fertilizer $NH_4^+$-N treatment on the third day, but it rebounded from 4 to 7 days, mainly due to the $NH_4^+$-N accounts for 88.6% of the total nitrogen in the liquid digestate, and other nitrogen-containing organic substances in the liquid digestate components are oxidized and decomposed by microorganisms, which increases the $NH_4^+$-N content of the overlying water. The concentration of $NH_4^+$-N in overlying water treated with liquid digestate for 0 days (the sampling time in this study is within 8 h after application) is significantly higher than that of fertilizer $NH_4^+$-N treatment, but the concentration of $NH_4^+$-N in pore water and the concentration of $NH_4^+$-N adsorbed by soil are lower than that of fertilizer $NH_4^+$-N treatment. This is because there is a competitive and mutually exclusive relationship between other cations [2] and $NH_4^+$-N ions in liquid digestate, thus delaying the molecular diffusion rate of $NH_4^+$-N in pore water, it also reduces the dominant sorption of $NH_4^+$-N on soil particles [47]. The abnormal value of BSN1 treatment on the fourth day of application may be related to the operation error of the destructive test, and the value at this point can be regarded as the missing value. After 7 days of application, the $NO_3^-$-N concentration in overlying water, pore water, and soil with the liquid digestate $NH_4^+$-N treatment was significantly higher than that of the fertilizer $NH_4^+$-N treatment, which may be that the organic active substances in the liquid digestate components promoted the reproduction of nitrifying microorganisms [48], thus promoting the conversion of $NH_4^+$-N to $NO_3^-$-N.

### 4.3. Fate of Farmland Disposal Liquid Digestate $NH_4^+$-N

Using farmland to consume liquid digestate is a widely recognized and effective treatment method, and the environmental pollution risk related to this measure has always been the focus of attention [26–28]. The total nitrogen in liquid digestate is mainly $NH_4^+$-N. There is no clear report on the final whereabouts of $NH_4^+$-N when a large amount of liquid digestate is applied to farmland. Some scholars believe that liquid digestate $NH_4^+$-N might enter the underlying soil through leaching and then pollute the groundwater [46], but many experimental studies have shown that the application of liquid digestate in dryland [29], and in paddy fields [16,33] has not significantly increased the $NH_4^+$-N and $NO_3^-$-N in groundwater. The results of this study showed that the concentration of $NH_4^+$-N in the pore water of 0–10 cm soil layer reached the peak value after 4 days of liquid

digestate application, and the $NH_4^+$-N sorption by soil particles reached the peak after 7 days, indicating that the $NH_4^+$-N in overlying water was gradually migrating to the soil layer over time. The proportion of liquid digestate nitrogen (including $NH_4^+$-N, $NO_3^-$-N transformed from $NH_4^+$-N) absorbed by soil and contained in pore water in the 0–10 cm soil layer accounted for 76.3% of the total $NH_4^+$-N after being applied for 15 days, indicating that the disposal of liquid digestate $NH_4^+$-N in farmland was mainly soil sorption and transformation. However, with the increase in liquid digestate application, the proportion of $NH_4^+$-N adsorbed by soil in the 0–10 cm soil layer decreases, which is due to the limit value of soil sorption capacity of 1108.55 mg/kg [49]. When the sorption limit value is exceeded, $NH_4^+$-N will migrate to the 10–20 cm soil layer. Only when the amount of $NH_4^+$-N applied exceeds the sorption limit value of 0–20 cm cultivated soil layer, there will be the risk of polluting groundwater.

Ammonia volatilization loss was once considered as one of the main ways to reduce $NH_4^+$-N in field water [31,34], but the ammonia volatilization process is very complex and affected by many factors, so it is very difficult to accurately estimate its loss under natural conditions. Some studies have shown that soil wetting or flooding conditions can reduce ammonia volatilization [37,38]. The standing test of liquid digestate showed that the removal rate of ammonium nitrogen was only 53% under natural conditions for 100 days [32]. In this study, the other fate of liquid digestate $NH_4^+$-N in the BSN1, BSN1.5, and BSN2.5 treatments accounts for 22.8%, 31.8%, and 34.1%, including the migration of $NH_4^+$-N to deeper soil layer [23,24], transformation [25] and ammonia volatilization loss of overlying water. Some scholars believe that the key period of ammonia volatilization is within 7 days after liquid digestate application. The ammonia volatilization loss rates of 1 N liquid digestate, 2 N liquid digestate, 4 N liquid digestate, and 1 N chemical fertilizer treatment are 18.8%, 14.3%, 9.9%, and 6.6%, respectively [32]. In this experiment, the other directions of liquid digestate $NH_4^+$-N were not subdivided, so the proportion of ammonia volatilization loss in other directions could not be determined. Therefore, it cannot be proved that ammonia volatilization loss is one of the main ways to reduce liquid digestate $NH_4^+$-N in overlying water. However, at the beginning of liquid digestate application, the concentration of $NH_4^+$-N in overlying water maintained a high level, decreased rapidly within 7 days, and gradually transformed into $NO_3^-$-N after 7 days. Therefore, it is speculated that if ammonia volatilization really exists, then the critical period of ammonia volatilization should be within 7 days, but the proportion of ammonia volatilization loss will not be too high.

Limitations and Directions for Future Research

In this study, only a typical pig farm liquid digestate and a typical farmland soil were selected. Whether the research results have universality for different types of liquid digestate and soil needs further research.

## 5. Conclusions

The use of saturated water content farmland soil for disposal of liquid digestate ammonium nitrogen is mainly based on soil sorption and pore water migration. With the extension of time, the ammonium nitrogen concentration in the overlying water gradually decreases, and the reduction rate of the total ammonium nitrogen gradually increases. However, the reduction speed and reduction rate showed a downward trend with the increase in the application amount of the ammonium nitrogen in the liquid digestate. The application of 0–3 d is the rapid consumption period for preventing and controlling the pollution of surrounding water bodies, and the application of 15 d is the completion period of one-time safety consumption.

**Author Contributions:** Conceptualization, Z.W. and J.S.; methodology, Z.W.; software, Z.W.; validation, X.Y., A.O.O. and E.B.G.K.; formal analysis, J.S.; investigation, X.Y.; resources, L.Z.; data curation, Z.W. and G.S.; writing—original draft preparation, Z.W.; writing—review and editing, Z.W. and

E.B.G.K.; supervision and revision, A.O.O. and H.S.; project administration, W.Z.; funding acquisition, Z.W. All authors have read and agreed to the published version of the manuscript.

**Funding:** This research was funded by the National Key Research and Development Program of China (2018YFD0800105), the Jiangsu Agricultural Science and Technology Innovation Fund of China (CX(20)2014), the Jiangsu Province Key Research and Development Project (Modern Agriculture) of China (BE2019395), and the Collaborative Extension Plan of Jiangsu Major Agricultural Technologies of China (2020-SJ-047-04-01).

**Data Availability Statement:** The data presented in this study are available on request from the first author and the corresponding author.

**Acknowledgments:** Key Laboratory of Crop and Livestock Integrated Farming is thanked for the support provided in soil and water determinations. Jiangsu Agricultural Technology Extension Station is thanked for the economic support provided for publication fee.

**Conflicts of Interest:** The authors declare no conflict of interest.

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
