# Peer review of "Aging Characteristics and Fate Analysis of Liquid Digestate Ammonium Nitrogen Disposal in Farmland Soil"

_water, doi:10.3390/w14162487_

Round 1

Reviewer 1 Report

The manuscript entitled “Aging Characteristics and Fate Analysis of Digesting Biogas Slurry Ammonium Nitrogen in Farmland Soil” is basically a case of study focused on fate of the ammonia during the application of a digestate on a farmland soil. The authors must address two issues before the article can be published:

-        -  In the section 4 and 5, the Authors highlight the good results they obtained (it is ok!), but they also should highlight that their results come from an experimental campaign characterized by one kind of digestate and one kind of soil. It is well known that the characteristics of the digestate produced by an anaerobic digester could change daily (seasonality, feed, OLR, and so on…). Let imagine the variations in the characteristics of the digestates from different digesters! The same is for the farmland soil. Consequently, are the results obtained under the conditions adopted by the authors reproducible with other digestates and/or soils? This topic has to be addressed by the Authors in section 4 and 5.

-        -  Be careful with the adoption of term “adsorption”. If the Authors use the this word, they have to prove that the physical phenomenon occurs. Do the Authors have the evidence of this phenomenon? Otherwise, they should use the term sorption. In this regard, I strongly suggest the reading of this recent and interesting paper: Pourret, O., Bollinger, J. C., Hursthouse, A., & van Hullebusch, E. D. (2022). Sorption vs Adsorption: the words they are a-changin', not the phenomena. Science of The Total Environment, 156545.

Other minor revisions:

-        -  Please, enlarge figures and legends.

-        -  Line 31: to improve the environmental pollution? I believe (and I hope) that the goal is to reduce environmental pollution.

-       -   I disagree with the use of “biogas slurry” instead of “digestate”.

-       -   Section 4.1 should be rewritten. Some sentences, such as “After the application of 1–2 times biogas slurry nitrogen for 3 days”, “After biogas slurry application 3 days”, “After application 7 days”, cause serious misunderstandings

Author Response

We greatly appreciate your positive comments and insightful suggestions.

We have provide a point-by-point response to your comments.

Reviewer 2 Report

In this paper, simulation experiments of disposing liquid digestate (with application of 0, 120, 180, and 300 kg NH4+-N /hm2) by static soil column are carried out in order to study the treatment efficiency, migration and transformation characteristics, and fate proportion of NH4+-N in saturated water content soil. Some significant data have been obtained, which can provide reference for the utilization of digestate. However, there are big problems with the paper.

Here are the problems.

1. “biogas slurry” should be “liquid digestate”.

2. Line 103. “black film sealed storage” should be “covered lagoon”.

3. Section 2.2. Static Soil Column should be given a schematic diagram.

4. Line 162. What were the overlying water? liquid digestate did not infiltrate after liquid digestate was applied to Static Soil Column?

5. Line 227. If liquid digestate is applied to farmland, the overlying water discharge into water body should not meet the emission standard of pollutants for livestock and poultry breeding industry (GB18596-2001).

6. Line 203. How to measure V ?

7. Line 315. “digestion capacity”should be “bearing capacity”.

8. What is the purpose of the paper? The main purpose of liquid digestate application in farmland soil is to provide fertilizer for crops, not to treat liquid digestate for meeting discharge standard. The treatment of liquid digestate for meeting discharge standard should use aerobic process or rapid filter-land treatment system.

9. There are many problems with language and terminology.

Author Response

(The authors gave the same response as above.)

Reviewer 3 Report

The review report

The peer-reviewed article, titled “Aging Characteristics and Fate Analysis of Digesting Biogas Slurry Ammonium Nitrogen in Farmland Soil”, addresses the important problem of using the by-product of anaerobic digestion for rational soil fertilization.

The topic presented in the article is currently widely researched in the world. The research in the manuscript was well prepared, however, in the article, I found a number of ambiguities, which I listed in the detailed comments below.

Introduction section: The authors use the term biogas slurry, could they define it more precisely? I am a little confused because biogas slurry (digestate) is a liquid containing water + solid particles. Most biogas plants use the separators or even centrifugal cleaners to separate the solid matter from slurry. Can Authors explain, does the biogas slurry is after solid matter separation?

I suggest using the name of raw digestate, not "biogas slurry". Or explain the differences between digestate and biogas slurry. Typically, the digestate from the bioreactor is further separated into the solid and the liquid fractions. The title of the article can be misleading because the reader may not know whether the studies used solids or soluble substances contained in the biogas plant by-product.

Please highlight the novelty in the presented research, what new knowledge can be provided by research. The digestate utilization studies have already been presented in numerous studies, such as: 10.1016 / j.envpol.2017.05.023.

Section 2.1: For me, it is unclear here, it means that only wastewater with soluble components was used for further research. Were solid, insoluble matter filtered out of the digestate and not used for the experiment? Is it true? Please clearly mark such information in the manuscript. The title of the article should also be modified by adding the information that the effect of soluble fractions in digestate is being investigated.

Section 2.3: I suggest using the SI units, ie mg / dm3 instead of mg / L.

Section 2.4: Samples were taken 9 times, three samples each time. The authors found that 30 measurements were taken for each treatment, but simple multiplication indicates a different number of repeats 9x3 = 28. Was there a calculation error here?

… full-automatic flow analyzer (SKALAR SAN + +): Please add the manufacturer name, (Skalar Analytical B.V. Products, Netherlands).

Section 4.3, line 393: Please specify here what is the limit value of the adsorption.

The reviewed article is interesting. The research work has been well planned and the assumptions are complementary. However, it has many gaps and inaccuracies, which should be completed before going to print.

Author Response

(The authors gave the same response as above.)

Reviewer 4 Report

The article is scientifically correct, well-described methodology and research results. In addition to the scientific influence, it is of great practical importance not only locally.

Author Response

We greatly appreciate your positive comments. 

Round 2

Reviewer 1 Report

Now the manuscript is worth of publication

Author Response

We thank you again for your understanding and support of our manuscript.

Reviewer 2 Report

The author does not wish to revise "biogas Slurry" to "liquid digestate".However, no reasonable explanation was given. The residue after anaerobic digestion is often referred to as "digestate". I still suggest to revise "biogas Slurry" to "liquid digestate".

Author Response

We thank you again for your insightful suggestion. We revised as suggested. (in red)

Reviewer 3 Report

Dear Authors,

Thank you for the corrections made to the manuscript. Most of my comments were taken into account, and the doubts concerning the presented research were cleared up. The justification for not changing the mg / L unit raises doubts. Using such a unit mg / L instead of mg / dm3 is actually accepted in the technical nomenclature. In my opinion, it should not be used for scientific publications. It is nevertheless rather a technical note. In my opinion the manuscript can be published.

Best regards

Reviewer

Author Response

We thank you again for your  insightful suggestions and support of our manuscript.